# Polyphasic Approach Utilized for the Identification of Two New Toxigenic Members of *Penicillium* Section *Exilicaulis*, *P. krskae* and *P. silybi* spp. nov.

**DOI:** 10.3390/jof7070557

**Published:** 2021-07-13

**Authors:** Roman Labuda, Markus Bacher, Thomas Rosenau, Erika Gasparotto, Hannes Gratzl, Maria Doppler, Michael Sulyok, Alena Kubátová, Harald Berger, Kristof Cank, Huzefa A. Raja, Nicholas H. Oberlies, Christoph Schüller, Joseph Strauss

**Affiliations:** 1Unit of Food Microbiology, Department for Farm Animals and Veterinary Public Health, Institute of Food Safety, Food Technology and Veterinary Public Health, University of Veterinary Medicine Vienna, Veterinaerplatz 1, 1210 Vienna, Austria; 2Research Platform Bioactive Microbial Metabolites (BiMM), Konrad Lorenz Strasse 24, 3430 Tulln a.d. Donau, Austria; erika.gasparotto@boku.ac.at (E.G.); joseph.strauss@boku.ac.at (J.S.); 3Department of Chemistry, University of Natural Resources and Life Sciences, Vienna (BOKU), Konrad Lorenz Strasse 24, 3430 Tulln a.d. Donau, Austria; markus.bacher@boku.ac.at (M.B.); thomas.rosenau@boku.ac.at (T.R.); 4Institute of Bioanalytics and Agro-Metabolomics Department of Agrobiotechnology (IFA-Tulln), University of Natural Resources and Life Sciences, Vienna (BOKU), Konrad Lorenz Strasse 20, 3430 Tulln a.d. Donau, Austria; hannes.gratzl@boku.ac.at (H.G.); maria.doppler@boku.ac.at (M.D.); michael.sulyok@boku.ac.at (M.S.); 5Department of Botany, Faculty of Science, Culture Collection of Fungi (CCF) Charles University, Benátská 2, 128 01 Prague, Czech Republic; alena.kubatova@natur.cuni.cz; 6Fungal Genetics and Genomics Laboratory, Department of Applied Genetics and Cell Biology, Institute of Microbial Genetics, University of Natural Resurces and Life Sciences, Vienna (BOKU), Konrad Lorenz Strasse 24, 3430 Tulln a.d. Donau, Austria; harald.berger@boku.ac.at (H.B.); christoph.schueller@boku.ac.at (C.S.); 7Department of Chemistry and Biochemistry, University of North Carolina at Greensboro, Greensboro, North Carolina 27402, USA; k_cank@uncg.edu (K.C.); haraja@uncg.edu (H.A.R.); n_oberli@uncg.edu (N.H.O.); 8Core Facility Bioactive Molecules: Screening and Analysis, University of Natural Resources and Life Sciences, Konrad Lorenz Strasse 24, 3430 Tulln a.d. Donau, Austria

**Keywords:** cycloheximide tolerance, *Penicillium restrictum* group, milk thistle, red exudate, secondary metabolites, mycotoxins

## Abstract

Two new species, *Penicillium krskae* (isolated from the air as a lab contaminant in Tulln (Austria, EU)) and *Penicillium silybi* (isolated as an endophyte from asymptomatic milk thistle (*Silybum marianum*) stems from Josephine County (Oregon, USA)) are described. The new taxa are well supported by phenotypic (especially conidial ornamentation under SEM, production of red exudate and red pigments), physiological (growth at 37 °C, response to cycloheximide and CREA), chemotaxonomic (production of specific extrolites), and multilocus phylogenetic analysis using RNA-polymerase II second largest subunit (*RPB2*), partial tubulin (*benA*), and calmodulin (*CaM*). Both new taxa are resolved within the section *Exilicaulis* in series *Restricta* and show phylogenetic affiliation to *P. restrictum sensu stricto*. They produce a large spectrum of toxic anthraquinoid pigments, namely, monomeric anthraquinones related to emodic and chloremodic acids and other interesting bioactive extrolites (i.e., endocrocin, paxilline, pestalotin, and 7-hydroxypestalotin). Of note, two bianthraquinones (i.e., skyrin and oxyskyrin) were detected in a culture extract of *P. silybi*. Two new chloroemodic acid derivatives (2-chloro-isorhodoptilometrin and 2-chloro-desmethyldermoquinone) isolated from the exudate of *P. krskae* ex-type culture were analyzed by nuclear magnetic resonance (NMR) and liquid chromatography–mass spectrometry (LC–MS).

## 1. Introduction

*Penicillium* Link (Aspergillaceae, Eurotiales, Ascomycota) is a diverse genus with more than 483 species [1]. Members of this genus are very rich in the production of secondary metabolites [2,3,4] and are capable of producing an exceptionally diverse range of bioactive compounds with important biotechnological applications. A large number of *Penicillium* species have been isolated from a myriad of ecological habitats and are studied for their taxonomic and chemical diversity [5]. *Penicillium* species are known for their contributions to secondary metabolites of pharmaceutical applications such as antibiotics, cholesterol-reducing agents, and stains. A number of these species are known to produce secondary metabolites known as mycotoxins. Although such fungi produce an enormous array of secondary metabolites, the term mycotoxins is restricted to those secondary metabolites that pose a potential health risk to animals and humans exposed to these natural products through contamination of feed and food [6,7]. The vast majority of toxigenic species within the genus is associated with agricultural crops, such as cereals, fruits, and vegetables [8], belonging to a group of so-called terverticillate penicillia that are currently affiliated into subgenus *Penicillium* [9].

The species outside the subgenus *Penicillium* are currently affiliated to subgenus *Aspergilloides* [9], divided into section *Aspergilloides* [10] and section *Exilicaulis* [11]. These two sections are microscopically characterized with mono- and biverticillate conidiophores, respectively. In general, penicillia of the subgenus *Aspergilloides* are primarily soil-borne fungi [12,13,14] but can play important roles in a variety of habitats, such as causing allergies in indoor environments or being reported from human and veterinary clinical cases [11,15,16]. Others have been found to be significant in deterioration of stored rice or other crops and producing toxic secondary metabolites (i.e., mycotoxins), and thus they can represent a health risk for consumers exposed to contaminated food or feeds [8,11]. A typical example is *P. citreonigrum*, which produces the neurotoxin citreoviridin [4,8].

The mycotoxin producing clade *Penicillium* sect. *Exilicaulis*, typified by *P. restrictum*, is characterized by monoverticillate conidiophores and non-vesiculated stipes [9,11]. The sect. *Exilicaulis* has recently been studied intensively with the use of Genealogical Concordance Phylogenetic Species Recognition (GCPSR). A number of studies that have focused on resolving taxonomic and phylogenetic delimitation of *Penicillium* species have used a polyphasic approach such as GCPSR and extrolites examination to identify unknown fungal cultures to novel species in the different sections of the genus. As numerous new species were described in this section [11], it seems likely that more species in this clade await discovery.

During routine microbiological work in our laboratory (revitalization of the internal BiMM collection fungal strains), a slow growing *Penicillium* with simple monoverticillate conidiophores producing copious ruby-red exudate was found on a malt extract agar Petri plate in July 2019 as a fungal contamination. This isolate was designated as BiMM-F280. A morphologically similar fungus identified as *Penicillium restrictum*, strain G85 producing red guttates, was isolated as an endophyte from *Silybum marianum* (milk thistle) stem [17,18]. To place these fungi into a phylogenetic framework, we obtained sequences from four loci, recommended for *Penicillium* taxonomy [19], i.e., internal transcribed spacer region including 5.8S rDNA (ITS), RNA-polymerase II second largest subunit (*RPB2*), partial tubulin (*BenA*), and calmodulin (*CaM*). In addition, we also examined the secondary metabolite production of these strains. Overall, the resulting data, combining morphology, physiology, and molecular phylogeny, revealed that both strains (BiMM-F280 and G85) represent novel species of the genus *Penicillium* Link, section *Exilicaulis*, in the *P. restrictum* clade.

## 2. Material and Methods

### 2.1. Sample Collection and Isolation of the Fungi

A single colony of a red-pigmented fungus (BiMM-F280) was observed during routine mycological work (during revitalization of fungal strains), appearing on a malt extract plate in July 2019 in our laboratory (Tulln, Austria). The strain G85 was isolated as an endophyte from asymptomatic milk thistle (*Silybum marianum* (L.) Gaertn. (Asteraceae)) stems in USA, Oregon, Horizon Herbs, Josephine County (N 42 8.639 W 123 17.595), on 19 September 2011 [17,18,19].

### 2.2. Cultivation of the Strains, Media, and Morphological Analysis

For phenotypic characterization, the strains were transferred to Czapek yeast extract agar (CYA), malt extract agar (MEA), yeast extract sucrose agar (YES), and creatine sucrose agar (CREA), as previously described [20], as well as on potato dextrose agar (PDA, Fluka) Sabouraud 4% dextrose agar (SDA, VWR), and incubated for 7–30 days in the dark at 25 °C. Colony size (in mm), structure, pigmentation, and characteristics were recorded after 7 and 30 days. The cultivation was prolonged up to 30 days to determine the onset of sexual reproduction organs. To determine the optimal and minimal/maximal temperatures for growth, we incubated the strains on CYA and MEA at 5, 9, 12, 20, 25, 30, 37, and 39 °C (±0.1–0.2 °C) for seven days. Malt extract agar with added cycloheximide (Sigma), a protein synthesis inhibitor at concentrations of 100, 250, and 500 ppm (µg.ml^−1^) (MEA-CX100, MEA-CX250, and MEA-CX500, respectively), was used to determine in vitro susceptibility. For comparative descriptions of the macroscopic and microscopic characteristics, CYA and MEA were used [10,11,21,22].

For determination of microscopic traits, MEA was utilized, and cultures were allowed to grow for 7–9 days. Conidiophore and conidial formation were observed in situ under low magnification (50 – 100×). Details of conidiophores, conidia, and other microscopic structures, such as width of hyphae, were observed in mounts with lactophenol blue (RAL Diagnostics). For hyphae width, 25 measurements were made and represented as maximal value in micrometers. For conidiophore, branch and phialide length and width 30 measurements were recorded, and the data are represented as (minimal-), typical range, and (-maximal) value in micrometers. For conidia, 50 measurements were made, and the data are represented as (minimal-), typical range, and (-maximal) value, including a mean and standard deviation. Scanning electron microscopy (SEM) was performed on a JEOL-6380 LV microscope (JEOL Ltd. Tokyo, Japan). Pieces of colonies (5 × 3 mm) that were grown 3–4 weeks on MEA and OA were fixed in osmium tetroxide vapors 1 week at 5–10 °C and then were gold-coated in Bal-Tec SCD 050 sputter coater. The specimens were observed with spot size 42 mm and accelerating voltage 20 kV. The surface ornamentation of the conidia (morphs) was classified into four categories (tuberculate, aculeate-echinulate, verrucose, and lobate-reticulate) [23].

Dried herbarium specimens of the holotypes were deposited in the herbarium of the Mycological Department, National Museum in Prague, Czech Republic (PRM); the ex-type cultures were deposited in the Bioactive Microbial Metabolites (BiMM) Fungal Collection, UFT - Tulln (AT), and in the Culture Collection of Fungi (CCF), Prague (CZ).

Photomicrographs were captured using phase and Nomarski contrast on an Olympus CellR BX51 microscope with Olympus DP72 camera and QuickPHOTO Micro 3.0 software. Melzer’s reagent and lactic acid with cotton blue were used as a mounting medium for microphotography. Photographs of the colonies were taken with a Sony DSC-RX100.

### 2.3. DNA Extraction, PCR Amplification, and Sequencing

DNA was extracted from the strains grown on MEA for seven days using the DNeasy Plant Minikit (Qiagen, Germany). Amplification of the ITS, RPB2, BenA, and CaM were performed as previously described [19].

Regions of internal transcribed spacer (ITS), β-tubuline (BenA), calmodulin (CaM), and RNA-polymerase II second largest subunit (RPB2) were amplified using primer pair ITS1 and ITS4, Bt2a and Bt2b, CMD5 and CMD6, and 5Feur and 7CReur, respectively (19). The PCR thermal cycle programs used for amplification were those as described in (19).

All reactions were performed in an Eppendorf Gradient *MasterCycler* (Eppendorf, Hamburg). Conditions for amplification of ITS: 95 °C for 2 min and 30 s (initial denaturation), 35 cycles of 94 °C for 30 s (denaturation), 54 °C for 30 s (annealing), 72 °C for 1 min (elongation), and 72 °C for 5 min (final elongation); *BenA*: 95 °C for 2 min and 30 s, 35 cycles of 94 °C for 20 s, 54 °C for 20 s, 72 °C for 1 min, and a final elongation step at 72 °C for 5 min; *CaM*: 94 °C for 5 min; 35 cycles of 94 °C for 45 s, 55 °C for 45 s, 72 °C for 1 min, and a final elongation step at 72 °C for 7 min; *RPB2*: 94 °C for 5 min; 30 cycles of 45 s at 94 °C, 50 °C for 45 s, and a final elongation step at 72 °C for 7 min.

The PCR purification was performed using a Monarch PCR & DNA Cleanup kit (5 μg) (New England Biolabs, USA). Sanger sequencing was performed by LGC Genomics (Germany). All sequences obtained in this study were deposited in GenBank (Table 1).

### 2.4. Phylogenetic Analysis

Nucleotide sequences from 17 *Penicillium* species (Table 1) for the genes BenA, CaM, and RPB2 were aligned using ClustalW (Gap Opening Penalty 15.00, Gap Extension Penalty 6.66) within MEGA X (v. 10.1.8) [24]. Non-aligned sequence overhangs were removed, and the 3 gene alignments were concatenated into a single alignment using the R package Biostrings [25], which was used for phylogenetic distance estimation.

#### 2.4.1. Evolutionary Analysis Using Maximum Likelihood

The evolutionary history was inferred by using the maximum likelihood method and Jukes–Cantor model [26]. The percentage of trees (1000 bootstrap replicates) in which the associated taxa clustered together is shown next to the branches. Initial tree(s) for the heuristic search were obtained by applying the neighbor-joining method to a matrix of pairwise distances estimated using the Jukes–Cantor model. The tree is drawn to scale, with branch lengths measured in the number of substitutions per site. This analysis involved 17 nucleotide sequences. There were a total of 1676 positions in the final dataset. Evolutionary analyses were conducted in MEGA X [24]. *Penicillium corylophilum* ex-type culture CBS 312.48 was selected as an outgroup for phylogenetic evaluation.

#### 2.4.2. Bayesian Analysis

Bayesian inference phylogenies were inferred using MrBayes 3.2.6 [27], under partition model (2 parallel runs, 10 million generations), using PhyloSuite v. 2.1 [28]. Four independent chains of Metropolis-coupled MCMC were run for 10 million generations with trees sampled every 1000th generation, resulting in 10,000 total trees. By observing the average standard deviation of split frequencies value approaching 0.01, we estimated that the two runs had converged closer to stationary phase (10 million generations). Consensus trees were generated and viewed in PAUP* v.4.0a (build 166) [29]. Clades with a posterior probability (PP) ≥ 95% were considered significant and strongly supported.

### 2.5. Chemical Analysis of the Red Exudate and Metabolic Profiling

#### 2.5.1. Fermentation and Extraction

*A large-scale production of the red exudate:* The fungal spore suspension (5.0 × 10^6^ spores/mL) was obtained after 7 days of cultivation of the fungus (*P. krskae* BiMM-F280) on a malt extract agar (MEA, [8]). Five colony plugs were cut (each at around 1 × 1 cm) and thoroughly mixed (on vortex for 2 min) with 30 mL physiological solution (0.9% NaCl) in a sterile, 50 mL capacity Falcon tube. A total of 2 L malt extract agar medium (MEA) spread over approximately 80 Petri plates was used for production of the red exudate. Each plate was inoculated with 100 µL spore suspension in the three parallel streaks at the central and sub-central part of the plate. The plates were cultivated in perforated plastic bags for 14 days, at 25 °C, in the dark. At the end of the cultivation, the plates were verified for the presence of contamination, and around 70 mL exudate droplets were collected manually with a pipette. The exudate was then directly mixed with 3 g of silica gel and dried under a blowing air for 24 h before separation via reverse phase flash chromatography. A small aliquot (50 µL) of the exudate was retained for a direct examination by LC–MS.

#### 2.5.2. Micro-Extraction of YES Cultures for Metabolic Profiling and Quantification

A total of 7–9 agar plugs (1 g), were taken off by the cork borer (9 mm in diameter) from a culture of *Penicillium* growing on YES plate for 14 days at 25 °C, in the dark. Then, it was transferred into an 8 mL capacity screw vial and mixed with 3 mL of acetonitrile/water/acetic acid 79:20:1 (*v*/*v*/*v*). After soaking (for 10 min) and vigorous vortexing (three times for 2 min), the extract was centrifuged (1200 rpm, at 10 °C, for 15 min) and filtrated through 0.2 µm Ø filter before LC–MS analysis. The blank sample was prepared in the same way by extraction of 1 g pure YES plate.

#### 2.5.3. Isolation of Secondary Metabolites from Red Exudate

The crude pigment (dried exudate sample) was applied to a reversed-phase silica gel vacuum flash chromatograph (Interchim, puriFlash^®^450), using two superimposed Interchim puriFlash^®^ 32 g silica IR-50C18-F0025 flash columns (particle size: 50 µm). The columns were eluted with a two-solvent gradient (solvent A: H_2_O, solvent B: CH_3_CN). The starting linear gradient of AB from 10% B to 27% B in 25 min at a flow rate 15 mL/min was followed by an isocratic gradient of AB at 52% B for 10 min, and then by a linear gradient from 52% to 66% B in 7 min at the same flow rate, and finally the column was washed starting with 100% B for 10 min followed by 100% A for 10 min at a flow rate 15–30 mL/min. UV 254 nm and UV scan 200–400 nm mode were used for detection and final separation of 5 main peak fractions (F1-F5), which were consequently concentrated under reduced pressure at 45°C. The target compounds were found in fraction F1 and F2 (yield: ≈100 mg) and resolved in solvent mix (1:1:1; CH_3_CN/CH_3_OH/H_2_O) and further purified by an Agilent 1260 Infinity preparative HPLC (USA) on a reversed phase column Gemini NX C-18 (21.20 × 150 mm, 5 µm, 110 Å), using a gradient that started with 15% to 95% CH_3_CN/H_2_0 in 30 min and a flow rate of 25 mL/min. After one stage of prep HPLC, a total of four preparative fractions were collected with yields (1, tR 14.35 min) 1.61 mg, (2, tR 15.95 min) 5.78 mg, (3, tR 17.5 min) 15.4 mg, and (4, tR 18.43 min) 8.59 mg. For purity check, an Agilent 1200 system with the same setup of column and gradient was used as mentioned above. The retention times refer to analytical HPLC.

#### 2.5.4. LC–MS Analysis of the Exudate

The data were collected on QExactive Plus mass spectrometer (ThermoFisher, San Jose, CA, USA). The spray voltage was set to 3 kV, the nitrogen sheath gas was set to 47.5 arb, and the auxiliary gas was set to 11.25 arb in the method. The initial data were collected from *m/z* 125 to 2000 for both positive and negative switching. The QExactive Plus was coupled to an Acquity ultra-performance liquid chromatography (UPLC) system (Waters Corp., Milford, MA). For the analysis, the flow rate was 0.3 mL/min using BEH C18 column (2.1 × 50 mm × 1.7 μm) at 40 °C. Sample temperature was 10 °C. Over a period of 10 min, the analysis started at 15% CH_3_CN and increased linearly to 100% CH_3_CN over 8 min; it was then held at 100% CH_3_CN over 1.5 min before returning to the starting conditions over 0.1 min. The starting conditions were held for an additional 0.4 min before the method ended. Photodiode-array (PDA) detection was used to acquire data from 200 to 500 nm with a resolution of 4 nm.

#### 2.5.5. Multi-Toxin LC–MS/MS Method for the YES Extract Quantification

The samples derived from YES plates was followed by dilution of 1 + 9 in acetonitrile/water/acetic acid 49.5/49.5/1 (*v*/*v*/*v*). Further dilutions of 1 + 49 and 1 + 999 were performed and re-analyzed in cases of distorted peak shapes due to column overloading caused by large analyte concentrations. The method used in this study is an extension of the version described in detail elsewhere [30,31,32]. Briefly, a QTrap 5500 MS/MS system (Sciex, Foster City, CA, USA) equipped with a TurboV electrospray ionization (ESI) source was coupled to a 1290 series UHPLC system (Agilent Technologies, Waldbronn, Germany). Chromatographic separation was performed at 25 °C on a Gemini C18-column, 150 × 4.6 mm i.d., 5 μm particle size, equipped with a C18 security guard cartridge, 4 × 3 mm i.d. (both Phenomenex, Torrance, CA, USA). Two MS/MS transitions were acquired per analyte with the exception of moniliformin and 3-nitropropionic acid that yield only one product ion. For confirmation of a positive identification, the ion ratio had to agree with the related values of the related authentic standard within 30% as stated in official guidelines [32], whereas for the retention time, a more strict in-house criterion of ± 0.03 min was applied.

#### 2.5.6. NMR

All NMR spectra were recorded on a Bruker Avance II 400 (resonance frequencies 400.13 MHz for ^1^H and 100.63 MHz for ^13^C) equipped with a 5 mm N_2_-cooled cryo probe head (Prodigy) with z–gradients at room temperature with standard Bruker pulse programs. The sample was dissolved in 0.6 mL of MeOD (99.8 % D) + some drops of DMSO-d_6_ (99.8 % D). Chemical shifts are given in parts per million, referenced to residual solvent signals (3.31 ppm for ^1^H, 49.0 ppm for ^13^C). ^1^H NMR data were collected with 32 k complex data points and apodized with a Gaussian window function (lb = −0.3 Hz and gb = 0.3 Hz) prior to Fourier transformation. ^13^C spectrum with WALTZ16 ^1^H decoupling was acquired using 64 k data points. Signal-to-noise enhancement was achieved by multiplication of the FID with an exponential window function (lb = 1 Hz). All two-dimensional experiments were performed with 1 k × 256 data points, while the number of transients (2–16 scans) and the sweep widths were optimized individually. HSQC experiment was acquired using adiabatic pulse for inversion of ^13^C and GARP-sequence for broadband ^13^C-decoupling, optimized for ^1^*J*_(CH)_ = 145 Hz. For the NOESY spectrum, a mixing time of 0.8 s was used.

#### 2.5.7. Confirmation of Novel Compounds by LC–MS

The diluted and purified compounds were analyzed by LC–HRMS using a Vanquish LC system coupled to a QExactive HF (Thermo Fisher Scientific, Bremen, Germany). Chromatographic separation was carried out via C18-RP using a Gemini^®^, NX-C18 column (5µm, 110Å, 150 × 2 mm, Phenomenex, Torrance, CA, USA). The column temperature was 25 °C and the flow rate set to 300µL/min. Eluent A consisted of H_2_O + 0.1%FA, eluent B of MeOH +0.1%FA. Gradient elution was carried out after two initial minutes (15% B), increasing eluent B from 15 to 95% within 30 min. After constant 95% B for 3 min, the system was equilibrated with the starting conditions (15% B) for 10 min, resulting in total method duration of 45 min. Heated ESI was operated in polarity switching mode, and mass spectra were recorded in full scan and data dependent MS/MS mode. Full scan spectra were recorded with a resolution of 120,000 FWHM at *m/z* 200 from *m/z* 100 to 1000, MS/MS fragment spectra with FWHM 15,000 at *m/z* 200.

## 3. Results

### 3.1. Taxonomy

*Penicillium krskae* Labuda, Kubátová, C. Schüller & J. Strauss sp. nov.—Figure 1.

*MycoBank*: MB839112

*Etymology:* Latin, *krskae =* in honour of Professor Rudolf Krska, a leader of Institute of Bioanalytics and Agro-Metabolomics, University of Natural Resources and Life Sciences, Vienna, an expert in mycotoxin analytics.

Colonies on CYA presented slow-to-moderate growth with 18–20 mm diameter, after seven days, at 25 °C, centrally slightly umbonate, radially furcate (sulcate), with white to creamy floccose aerial mycelium, none to only very poor grayish green sporulation, copious hyaline exudate present, reverse yellowish white, pigment absent even after prolonged incubation. Colonies on MEA presented moderate growth, 20–23 mm, after seven days, at 25 °C, centrally slightly umbonate, radially furcate, with white floccose aerial mycelium becoming more orange at colony margins, poor to moderate turquoise to blue-green sporulation (better sporulation at 30 °C), copious hyaline to deep-red (ruby-colored) exudate present, reverse yellow, orange-red pigment diffusing into the agar after prolonged incubation (10–12 d) coloring also the reverse into red shades. Colonies on PDA presented slow growth, 18–20 mm diameter, after seven days, at 25 °C, plane, with white, yellow to orange floccose aerial mycelium, moderate to good turquoise to blue-green sporulation, copious hyaline to deep-red (ruby-colored) exudate present, reverse yellow to yellowish orange, dark yellow pigment diffusing into the agar after prolonged incubation (10–12 d). Colonies on SDA presented moderate growth, 18–22 mm diameter, after seven days, at 25 °C, in nearly all aspects similar to those on MEA, including turquoise to blue-green sporulation and copious hyaline to deep-red (ruby-colored) exudate formation, reverse yellow with reddish center and conspicuously wrinkled with concentric rings. Colonies on YES presented moderate growth, 22–28 mm diameter, after seven days, at 25 °C, centrally slightly umbonate, radially deep furcate, with white to pinkish floccose aerial mycelium, moderate to good turquoise to blue-green sporulation, exudate absent, reverse yellow becoming greenish in age (after 14 days), pigment absent. Colonies on CREA presented poor growth, 12–14 mm diameter, with a weak acid formation under colony.

Growth at 39 °C (CYA, MEA= 0 mm in diameter, no spore germination), 37 °C (CYA= 8–10 mm diam., MEA= 2–4 mm diameter), at 35 °C (CYA= 15–18 mm in diameter, MEA= 15–17 mm in diameter), at 30 °C (CYA= 24–27 mm in diameter, MEA= 25–28 mm in diameter), at 20 °C (CYA= 14–16 mm in diameter, MEA= 15–17 mm in diameter), at 15 °C (CYA= 0 mm in diameter, MEA= 7–9 mm in diameter), at 10 °C (CYA= 0 mm in diameter, no spore germination, MEA= 1–2 mm in diameter), at 5 °C (CYA and MEA= 0 mm in diameter, no spore germination). Growth on MEA-CX100 after seven days of incubation (8 mm), MEA-CX250 (5 mm), and MEA-CX500 (0 mm, no spore germination).

Vegetative hyphae hyaline, smooth-walled, and septate, (1.5–)2.0–2.5(–3.5) µm wide. Conidiophores mostly short, (4.5–)10.0–18.0(–25.0), but also longer ones, up to 150(–200) µm present, stipe (1.2–)1.5–2.0(–3.5) µm wide, hyaline and smooth-walled, monoverticillate, high proportion of irregularly branched (divaricate-biverticillate) conidiophores present with branches often arising low on the conidiophores, main conidiophore stipe or a branch bearing 3–6 (–12) phialides (Figure 1c,d), non-vesiculate. Phialides typically flask-shaped, (4.4–)5.0–6.5(–7.1) × (1.8–)2.0–2.4(–2.6), with conspicuous necks. Conidia (Figure 1f,g) produced in short to long, tangled chains, smooth but mostly finely to conspicuously rough-walled (verrucose under SEM, Figure 1g), globose to subglobose, (1.7–)2.0–2.5(–2.7) µm (mean = 2.1 ± 0.2, *n* = 50). No sexual (teleomorphic) morph observed.

The following compounds are produced by the ex-type culture: monomeric anthraquinones: (+)-2’S-isorhodoptilometrin, 1’-hydroxy-2’-ketoisorhodoptilometrin, 7-chloro-1’-hydroxyisorhodoptilometrin, 1’-hydroxyisorhodoptilometrin, 2-chloroemodic acid, 2-chloro-isorhodoptilometrin [2-chloro-1,3,8-trihydroxy-6-(2-hydroxypropyl)anthracene-9,10-dione], 2-chloro-desmethyl dermoquinone [2-chloro-1,3,8-trihydroxy-6-(2-oxopropyl)anthracene-9,10-dione], 7-chlorocitreosein (chloro-citreorosein), O-demethyldermoquinone, emodic acid, emodin, ω-hydroxyemodin (citreorosein). Other extrolites: 7-hydroxypestalotin, NP1243, paxilline, and pestalotin.

Holotype: Austria, Tulln an der Donau, isolated as a (an air) contaminant of a malt extract agar plate, July 18 2019, isolated by Roman Labuda; PRM 955188 (Holotype, dried culture).

Ex-type culture: BiMM-F280 = CCF 6561= CBS 147776.

DNA sequences: GenBank MW794123 (ITS), MW774594 (*BenA*), MW774595 (*CaM*), and MW774593 (*RPB 2*).

Distinguishing characteristics: Presence of ruby exudate and/or pigment (on MEA, PDA, SDA, YES), relatively long monoverticillate conidiophores (up to 200 µm), smooth to rough conidia (verrucose under SEM), ability to growth at 37 °C, tolerance to cycloheximide (100–500 ppm), production of paxilline.

*Penicillium silybi* Labuda, Kubátová, Raja & Oberlies sp. nov.—Figure 2.

*MycoBank*: MB 839113

*Etymology:* Latin, *silybi =* referring to the host from which the fungus was isolated, *Silybum marianum* (L.) Gaertn. (Asteraceae) (i.e., milk thistle).

Colonies on CYA presented moderate growth, 26–28 mm diameter, after seven days, at 25 °C, centrally slightly umbonate, radially furcate (sulcate), with white floccose aerial mycelium, poor grayish sporulation, very minute hyaline exudate present, reverse creamy white, pigment absent. Colonies on MEA presented moderate growth, 28–30 mm diameter, after seven days, at 25 °C, similarly as CYA with a better gray sporulation, very minute hyaline exudate present, reverse yellow to dark yellow in the center, pigment absent. Colonies on PDA presented moderate growth, 30–32 mm diameter, after seven days, at 25 °C, plane and slightly umbonate at the center, with white floccose aerial mycelium, moderate gray sporulation, exudate absent (see notes), reverse lemon-yellow, bright yellow pigment present. Colonies on SDA presented moderate growth, 27–29 mm diameter, after seven days, at 25 °C, in nearly all aspects similar to PDA, except colonies are slightly depressed at the center and presence of hyaline to rose exudate, reverse lemon-yellow, bright yellow pigment (later red) present. Colonies on YES presented moderate growth, 25–27 mm diameter, after seven days, at 25 °C, centrally slightly umbonate and a crater forming, radially deep furcate, with white floccose aerial mycelium, poor gray sporulation, exudate absent, reverse bright yellow with orange center, pigment yellow orange (sometimes red after 14 days). Colonies on CREA presented poor and profuse growth, 20–22 mm diameter, with a moderate acid formation.

Growth at 39 °C (CYA, MEA= 0 mm in diameter, no spore germination), 37 °C (CYA= 3–5 mm diameter, MEA= 2–4 mm diameter), at 35 °C (CYA= 8–10 mm in diameter, MEA= 7–10 mm in diameter), at 30 °C (CYA= 25–27 mm in diameter, MEA= 25–27 mm in diameter), at 20 °C (CYA = 12–14 mm in diameter, MEA = 18–22 mm in diameter), at 15 °C (CYA = 5–7 mm in diameter, MEA= 8–10 mm in diameter), at 10 °C (CYA = 0 mm in diameter, no spore germination, MEA= 4–6 mm in diameter), at 5 °C (CYA= 0 mm in diameter, no spore germination, MEA = 0 mm in diameter, spore germination). Growth on MEA-CX100 after seven days of incubation (18 mm in diameter), MEA-CX250 (10 mm in diameter) and MEA-CX500 (5 mm in diameter). Minute droplets of ruby-red exudate present on all three MEA-CX after prolonged incubation (10–14 d).

Vegetative hyphae hyaline, smooth-walled and septate, (0.8–)1.5–2.5(–3.0) µm wide. Conidiophores short, up to (10–)15–18(–20) µm long, stipe (1.1–)1.4–1.6(–1.8) µm wide, hyaline and smooth-walled, monoverticillate, bearing 3–6(–12) phialides (Figure 2c,d), non-vesiculate. Phialides typically flask-shaped, (3.2–)4.0–5.0(–6.2) × (1.6–)2.0–2.5(–2.7), with conspicuous necks. Conidia (Figure 2e,f) produced in short to long, tangled chains, mostly rough-walled to spinulose and often striate (lobate-reticulate under SEM, Figure 2f), one-celled, globose (1.5–)1.8.0–2.2(–2.4) µm (mean = 2.0 ± 0.1, *n* = 50). No sexual (teleomorphic) morph observed.

The following compounds are produced by the ex-type culture: Monomeric anthraquinones: 2-hydroxyemodic acid, ( + )-2’S-isorhodoptilometrin, 1’-hydroxy-2’-ketoisorhodoptilometrin, 7-chloro-1’-hydroxyisorhodoptilometrin, 1’-hydroxyisorhodoptilometrin, 2-chloroemodic acid, 2-chloro-isorhodoptilometrin [2-chloro-1,3,8-trihydroxy-6-(2-hydroxypropyl)anthracene-9,10-dione], 2-chloro-desmethyl dermoquinone [2-chloro-1,3,8-trihydroxy-6-(2-oxopropyl)anthracene-9,10-dione], 7-chlorocitreosein (chloro-citreorosein), O-demethyldermoquinone, emodic acid, emodin, ω-hydroxyemodin (citreorosein), endocrocin. Bianthraquinones: oxyskyrin, skyrin. Other extrolites: 7-hydroxypestalotin, NP1243, pestalotin.

Holotype: USA, Oregon, 42°08′38.3″ N 123°17′35.7″ W, Josephine County, Horizon Herbs, isolated as an endophyte from asymptomatic milk thistle (*Silybum marianum*) stems, September 19 2011, isolated by Huzefa Raja; PRM 955189 (Holotype, dried culture).

Ex-type culture: G85 = CCF 6562 = CBS 147777.

DNA sequences: GenBank KF367458 (ITS), MW774592 (*BenA*), MW774591 (*CaM*), and AB860248 (*RPB 2*).

Distinguishing characteristics: Presence of ruby exudate and/or pigment (SDA, MEA-CX, sometimes on YES); short monoverticillate conidiophores (up to 20 µm); very rough to often striate conidia (lobate-reticulate under SEM); ability to grow at 37 °C; moderate growth at CREA with acid production; high tolerance to cycloheximide (100–500 ppm); production of endocrocin, skyrin, and oxyskyrin.

Notes: Strain G85 was originally isolated as a fungal endophyte from surface sterilized stems of milk thistle (*Silybum marianum*) [18]. A copious reddish exudate has been observed on MEA and PDA in a fresh isolate [17]

### 3.2. Phylogenetic Analysis

The phylogenetic tree constructed using the combined dataset *BenA*, *CaM*, and *RPB 2* (Figure 3) clearly indicates that the strains BiMM-F250 and G85 represent new species, named here as *P. krskae* sp. nov. and *P. silybi* sp. nov., respectively. The tree with the highest log likelihood (−4458.29) is shown.

*Penicillium krskae* is phylogenetically close to *P. kurssanovii* Chalab., and *P. silybi* to *P. chalabudae* Visagie. All four species cluster with *P. restrictum* J.C. Gilman & E.V. Abbott *sensu stricto* within series *Restricta* (genus *Penicillium*, subgenus *Aspergilloides*, section *Exilicaulis*). Specifically, series *Restricta* (formerly *P. restrictum*-clade) is composed of the eleven phylogenetically closely related species showing high bootstrap support (≥ 90%) and Bayesian posterior probabilities ≥ 95% (Figure 3).

### 3.3. Phenotype and Physiological Characteristics

The main distinguishing phenotypic characteristics used in this study with ex-type cultures of five phenotypically close species, namely, *P. krskae* sp. nov., *P. silybi* sp. nov., *P. chalabudae*, *P. kurssanovii*, and *P. restrictum*, could be categorized into five important taxonomic traits: (1) colony growth, (2) response to cycloheximide, (3) growth on CREA, (4) pigmentation (chemotaxonomically valuable character treated in detail in the chemical section), and (5) conidial morphology (Table 2).

(1) Growth conditions: At 37 °C, only *P. kurssanovii* did not present any growth (CYA, MEA). The slowest growth at this temperature was observed for *P. silybi* (3–5 mm in diameter on CYA), while *P. krskae*, *P. kurssanovii*, and *P. restrictum* showed moderately slow growth ranging from 8 to 10 (−12) mm in diam on CYA. At 25 °C, *P. silybi* presented the fastest growth on CYA (26–28 mm) and MEA (28–30 mm) after seven days when compared the other four species in this group, ranging from 12 to 20 mm (CYA) and 17 to 22 mm (MEA). At 15 °C, *P. krskae* presented no growth nor spore germination on CYA. The remaining four species presented minimal growth, ranging from 2 to 7 mm in diameter.

(2) Response to cycloheximide: Growth response of the new species and the phylogenetically close *P. chalabudae*, *P. kurssanovii*, and *P. restrictum* to 100, 250, and 500 ppm (µg.mL^−1^) cycloheximide in malt extract agar (Table 3) indicates a relatively high resistance of *P. krskae* and *P. silybi* to this fungicide. In particualr, *P. silybi* presented moderate growth, even at the highest concentration used (500 ppm).

(3) Growth and acid production on CREA: In general, all species showed only poor to slow growth on this medium with no to very weak acid production (*P. restrictum* and *P. kurssanovii*), a weak acid production under the colony only (*P. krskae*), or moderate acid production surrounding colonies (*P. silybi* and *P. chalabudae*).

(4) Exudate and pigment production: Red (ruby) exudate and/or pigment was observed only in the two new species, which correspond to metabolic profile and quantity of emodic acid derivatives (monomeric anthraquinones) found in these species as well as other important secondary bioactive extrolites produced by species within the investigated group of ex-type cultures (see metabolic profiling section).

(5) Conidia morphology: Detailed study of the morphology of conidia in terms of size and ornamentation revealed differences in an average size, ranging from 2.0 µm conidia of *P. silybi* to 2.4 µm conidia of *P. restrictum*. Spore surface ornamentation varied from verrucose (*P. krskae*, Figure 1g), lobate-reticulate (*P. silybi*, Figure 2f), tuberculate (*P. chalabudae*, Figure 4a, and *P. kurssanovii*, Figure 4b), and aculeate-echinulate (*P. restrictum*, Figure 4c).

### 3.4. Secondary Metabolites and Novel Chloroemodic Acid Derivatives

#### 3.4.1. Chemical Profiling of the Exudate

To access the chemical composition of the exudate of *P. krskae* and *P. silybi*, we grew the two fungi on two different media, PDA and MEA (Appendix A). The exudates were analyzed by LC–MS, and the identified metabolites were compared to an in-house database [30,33]. Thirteen different polyhydroxyanthraquinones (monomeric anthraquinones) were detected in the red exudates of *P. krskae* and *P. silybi* (Table 4), of which ten were previously reported to be produced by the exudate/guttates of these fungi [17]. The identification of these metabolites is considered to be first level of compound identification. In addition to accurate mass and retention time data, UV profiles were also used for comparison [34]. Two new chlorine-containing polyhydroxyanthraquinones, namely, 2-chloro-isorhodoptilometrin and 2-chloro-desmethyl dermoquinone, were identified for the first time from *P. krskae*.

#### 3.4.2. Metabolic Profile of the Related Penicillium Species in Series Restricta

The metabolic profile of the five closely related species (growing on YES medium for 14 days, at 25 °C, in the dark) without any conspicuous exudate production is highlighted in Table 5. The colonies, including the mycelium and agar, were extracted in a quantitative manner so that the detected compounds could be expressed as µg/g of fresh culture. As many as five extrolites, i.e., citreorosein, chlorocitreorosein, iso-rhodoptilometrin, metabolite NP1243, and roquefortine C, were found in all five ex-type cultures, with the largest production capacity being observed in *P. silybi* (≈1000-fold larger yields of the first three compounds compared to the capacity of other fungi tested). Alternatively, 15-hydroxyculmorin and hydroxysydonic and methylorsellinic acids were only found in *P. restrictum*, endocrocin and two bianthraquinones (oxyskyrin and skyrin), paxilline in *P. silybi*, and tryptophol in *P. chalabudae*.

#### 3.4.3. Two Novel Chloroemodic Acid Derivatives

Compounds **1** and **2** could only be isolated as a mixture with 2-chloroemodic acid (**3**) as its main component (**1**:**2**:**3** = 20:5:75). Therefore, structure elucidation was performed with this mixture. The ^1^H NMR signals for H-5 and H-7 of compound **3** were detected as doublets at δ_H_ 8.07 and 7.71 ppm, respectively, whereas the singlet of H-4 had a chemical shift of δ_H_ 7.23 ppm. For compounds **1** and **2**, both the resonances of H-5 and H-7 were shifted to higher fields by approximately 0.5 ppm (δ_H-5_ ≈7.50 and δ_H-7_ ≈7.20 ppm, respectively), whereas the singlets of H-4 remained nearly unchanged at δ_H_ 7.28 ppm. This indicated that the only structural difference to compound **3** would be in the side chain located on position C-6. For compound **1**, this side chain was elucidated as 2-hydroxypropyl group on the basis of a 3H doublet at δ_H_ 1.08 ppm, which showed a COSY crosspeak to an oxymethine proton at δ_H_/δ_C_ 3.88/66.6 ppm, which was further coupled to diastereotopic methylene protons at δ_H_ 2.69 and 2.76 ppm. These protons showed long-range crosspeaks in the HMBC spectrum to C-5 (δ_C_ 121.2) and C-7 (δ_C_ 125.0), as well as to C-6 (δ_C_ 150.5), finally proving this side chain being bound to C-6. LC–HRMS analysis of this fraction revealed a LC-peak with [M-H]^-^ peaks at *m/z* = 347.0329 and *m/z* = 349.0296 in the ratio of 3:1, typical for the presence of chlorine. On the basis of these data, we determined the molecular formula as C_17_H_13_ClO_6_ (calculated 347.0317), which confirmed the structure of **1** as presented in Figure 5. The side chain of compound **2** was identified as the corresponding 2′-keto derivative due to a methyl group (δ_H_/δ_C_ 2.19/30.1) and a methylene group (δ_H_/δ_C_ 3.99/49.2), both being singlets and showing long-range crosspeaks in the HMBC spectrum to a keto-carbon at δ_C_ 205.0 ppm. Additional crosspeaks from these methylene protons to C-5 (δ_C_ 121.5), C-7 (δ_C_ 125.6), and C-6 (δ_C_ 145.3) in combination with a molecular formula of C_17_H_11_ClO_6_ (calculated 345.0171) determined by [M-H]^-^ peaks in LC–HRMS spectra at *m/z* = 345.0172 and *m/z* = 347.0142 (ratio 3:1) confirmed structure **2** (Figure 5). Compounds **1** and **2** represent undescribed, new natural anthraquinone derivatives, although database search (SciFinder) showed that **1** is commercially available.

2-Chloro-isorhodoptilometrin [2-chloro-1,3,8-trihydroxy-6-(2-hydroxypropyl)anthracene-9,10-dione, **1**]: ^1^H NMR (DMSO-d_6_, 400 MHz, δ/ppm, *J*/Hz): 1.08 (d, 3H, 6.1 Hz, H-3’), 2.69 (dd, 1H, 13.2 + 7.4 Hz, H1’a), 2.76 (dd, 1H, 13.2 + 5.0 Hz, H-1’b), 3.89 (m, 1H, H-2’), 7.19 (d, 1H, 1.5 Hz, H-7), 7.28 (s, H-4), 7.54 (d, 1H, 1.5 Hz, H-5).

^13^C NMR (DMSO-d_6_, 100.6 MHz, δ/ppm): 23.6 (C-3’), 45.1 (C-1’), 66.6 (C-2’), 108.2 (C-4), 109.1 (C-9a), 112.9 (C-2), 113.8 (C-8a), 121.2 (C-5), 125.0 (C-7), 132.3 (C-4a/C-10a), 132.5 (C-4a/C-10a), 150.5 (C-6), 159.6 (C-1), 161.2 (C-8), 161.6 (C-3), 182.0 (C-10), 189.7 (C-9).

2-Chloro-desmethyl dermoquinone [2-chloro-1,3,8-trihydroxy-6-(2-oxopropyl)anthracene-9,10-dione, **2**]: ^1^H NMR (DMSO-d_6_, 400 MHz, δ/ppm, *J*/Hz): 2.19 (s, 3H, H-3’), 3.99 (s, 2H, H-1’), 7.17 (d, 1H, *J* = 1.5, H-7), 7.28 (s, 1H, H-4), 7.50 (d, 1H, *J* = 1.5, H-5).

^13^C NMR (DMSO-d_6_, 100.6 MHz, δ/ppm): 30.1 (C-3’), 49.2 (C-1’), 108.2 (C-4), 113.8 (-8a), 121.5 (C-5), 125.6 (C-7), 145.3 (C-6), 180.9 (C-10), 205.0 (C-2’).

## 4. Discussion

### 4.1. Phylogeny

The section *Exilicaulis* consists of six well-supported clades, where the monoverticillate species are resolved in the *P. citreonigrum*, *P. decumbens*, *P. parvum*, and *P. restrictum* clades, while the species with biverticillate conidiophores are clustered in the *P. melinii* and *P. corylophilum* clades [9,11]. These clades are currently treated as series: *Citreonigra*, *Alutacea*, *Erubescentia, Restricta*, *Lapidosa,* and *Corylophila*, respectively [1]. In our study, phylogenetic reconstruction applying combined sequence data of three loci—*RPB2*, *benA*, and *CaM*—resulted in clustering *P. krskae* sp. nov. with *P. kurssanovii* Chalab., and *P. silybi* sp. nov. with *P. chalabudae* Visagie along with *P. restrictum* J.C. Gilman & E.V. Abbott. Its sister clusters encompass six hitherto accepted species *P. arabicum* Baghd., *P. cinereoatrum* Chalab., *P. katangense* Stolk, *P. heteromorphum* H.Z. Kong & Z.T. Qi, *P. meridianum* D.B. Scott, and *P. philippinense* Udagawa & Y. Horie (Figure 3).

All aforementioned species are phylogenetically resolved within *Penicillium* section *Exilicaulis* in series *Restricta*, as was evident from the current study, as well as from the recent taxonomic revision [11], employing multigene phylogenies. The phylogenetically close species *P. kurssanovi*, *P. chalabudae, P. krskae*, and *P. silybi*, as well as a few others in the clade (i.e., *P. arabicum*, *P. katangense*), are thus far represented by only a single strain. However, the polyphasic approach used in this study clearly indicates that they are distinct species and they represent distinct taxa within the species concept currently adopted in series *Restricta* [1].

In the study of Visagie et al. (2016), the authors, recognized and accepted the following nine species *P. arabicum* (*syn. P. decumbens fide* Pitt 1980 [12]), *P. cinereoatrum*, *P. chalabudae* (= *P. albocinerascens*, *syn.*
*P. adametzii fide* Pitt 1980 [12]), *P. heteromorphum*, *P. katangense*, *P. kurssanovii* (*syn.*
*P. restrictum fide* Pitt 1980 [12]), *P. meridianum*, *P. philippinense*, and *P. restrictum sensu stricto* [11]. The authors of [11] noted that there is still a number of newly isolated strains from fynbos (South Africa) that may represent novel species. Thus, at the time of this study, series *Restricta* contains 11 well-defined species, including the two new taxa described in the present study.

### 4.2. Phenotype and Physiology

According to a traditional phenotype-mycological concept, Pitt (1980) [12], for instance, defined section *Exilicaulis*, with *Penicillium restrictum* as a type species, for those monoverticillate species (formerly subgenus *Aspergilloides*) in which the stipes do not produce a terminal vesicular swelling (i.e., with non-vesiculate stipes). Additionally, the author divided the section *Exilicaulis* into two series (*Restricta* and *Citreonigra*) on the basis of growth rates at 25 °C and stipe length, accepting 22 species. According to the new nomenclature, the genus *Penicillium* Link was redefined by Houbreaken et Samson (2011) [9] and later revised [10,11], and also Houbraken et al. (2020) revealed that 58 *Penicillium* spp. currently accepted and classified in section *Exilicaulis* [1], some of them with typically biverticillate conidiophores (formerly in subgenus *Furcatum*). On the basis of differences in phenotypic traits and secondary metabolites profiles, we can effectively separate the new described taxa from the phylogenetically closest species around *P. restrictum s. str.* within *Penicillium* section *Exilicaulis*.

A detailed study of phenotypical characteristics, especially growth at 37 °C, response to cycloheximide, growth and acid production on CREA, red exudate or pigment production, and conidial morphology (especially ornamentation), unambiguously confirms that the two new species are similar but not identical with their close relatives in series *Restricta*. Traditionally used physiological and morphological characteristics such as growth at different cultivation temperatures (especially on 15 and 37 °C), and media (such as CREA, CYA, MEA, and YES) and microscopy (including SEM) alone could provide a stable basis for accurate species identification. In addition, cycloheximide tolerance can be used as a stable diagnostic tool to distinguish phylogenetically closest taxa [35] on a generic level (a taxonomic character) with relatively constant responses of several *Penicillium* species [36]. Cycloheximide tolerance was also used for distinguishing non-toxigenic *Alternaria infectoria* from three closely related toxigenic *Alternaria* species [37]. However, this obviously important and consistent physiological character has not been firmly established in any other complex taxonomic study dealing with *Penicillium* and related genera [8,19].

The new species, *P. krskae*, is distinct compared to the phylogenetically close *P. kurssanovii* in many characters, especially (1) pigmentation (exudate and pigment) and coloration of mycelium (yellow-orange-pink vs. white); (2) conidial morphology (verrucose vs. tuberculate and an average size (2.1 vs. 2.3 µm); (3) degree of sporulation on CYA (poor vs. good); (4) growth at 37 °C on CYA (8–10 vs. 0 mm diameter); (5) acid production on CREA (present vs. absent to very weak); (6) response to cycloheximide differs in both species; and (7) a metabolic profile can be used to for distinguishing these two species, as *P. krskae* produces 7-hydroxypestalotin, pestalotin, and paxilline, while *P. kurssanovii* does not.

The second new species, *P. silybi*, can be distinguished from the phylogenetically closest *P. chalabudae* by the following traits: (1) response to cycloheximide at 250 ppm after 7 days of cultivation (growth vs. no spore germination); (2) generally slower growth at 35 and 37 °C; (3) colony characteristics on CREA; and (4) morphology of conidia, which are often striate in *P. silybi* (lobate-reticulate under SEM), while those of *P. chalabudae* are rough without striation and tuberculate under SEM. Although the two species share several metabolites, such as citreorosein and chlorocitreorosein, metabolites, such as emodin, endocrocin, oxyskyrin, and skyrin, were produced only by *P. silybi* in this study.

### 4.3. Extrolites

*Penicillium* species produce an exceptionally diverse range of secondary metabolites, and a number of these compounds have been shown to be moderately to highly potent toxins. However, the importance of particular *Penicillium* species as producers of mycotoxins depends as much on the ecology of the species as on the potency of the toxin produced [4].

Fungal guttation is a well-known phenomenon in the literature [35,38,39]. It has been suggested that these guttates or exudates play different ecological roles such as water reservoirs, excretion systems, or to degrade plant tissues [40,41,42]. Regardless of their role, guttates can be a rich source of secondary metabolites [17,43,44,45]. Due to the relatively small amount of material that these exudates provide, the isolation of secondary metabolites often occurs from the extract of the whole fungal culture rather than the guttate itself. In the present study, the chemical profiling of the exudates of *P. krskae* and *P. silybi* is presented. High-resolution LC–MS studies of the exudates revealed that the two fungi have nearly identical chemistry regardless of the media it was grown. All the previously reported metabolites, except 2-hydroxyemodic acid, were found to be produced by *P. krskae* and *P. silybi*. Furthermore, large-scale growth of the exudate of *P. krskae* was used to isolate the two novel chlorine-containing polyhydroxyanthraquinones, namely, 2-chloro-isorhodoptilometrin and 2-chloro-desmethyldermoquinone. Although 2-chloro-isorhodoptilometrin is a known compound in the database search (SciFinder), this is the first time it is described as a natural product. Overall, these results show that fungal exudates are chemically rich sources of secondary metabolites and that scale-up studies of these exudates can lead to the isolation of novel chemistry. Ongoing studies are examining their bioactivity against a panel of a microbial species.

From an organic extract of a solid phase rice culture of strain G85, researchers isolated a series of polyhydroxyanthroquinones [17]. Among these, ω-hydroxyemodin showed promising activity as a quorum-sensing inhibitor against clinical isolates of methicillin-resistant *Staphylococcus aureus* (MRSA) both in in vitro [17] and in vivo models [46]. Specifically, ω-hydroxyemodin (citreorosein) inhibited the MRSA response regulator AgrA and limits pathogenesis [46]. More recently, a scale up investigation of ω-hydroxyemodin resulted in improving and implementing both fermentation of fungal culture as well as isolation of pure compound to ≥ 800 mg in a laboratory setting [47]. Many anthraquinones exhibit antibacterial, antiparasitic, insecticidal, fungicidal, antiviral, and anticancer properties. However, at high concentrations, some anthraquinones may have toxic effects as mutagens and carcinogens [48]. Emodin is in nature the most common anthraquinone, having been isolated from fungi, lichens, flowering plants, and insects. It is an important intermediate in the biosynthesis of other fungal compounds [48].

Overall, the present study demonstrates that even well-explored taxa such as *Penicillium* are poorly known from a taxonomic and chemical perspective and can give rise to novel species and new secondary metabolites.

## 5. Conclusions

The combination of morphology, physiology, and molecular phylogeny; data; and the discovery of two new chloroemodic acid derivatives revealed that both strains described here represent novel species of the genus *Penicillium* Link, section *Exilicaulis*, in series *Restricta*.

## Figures and Tables

**Figure 1 jof-07-00557-f001:**
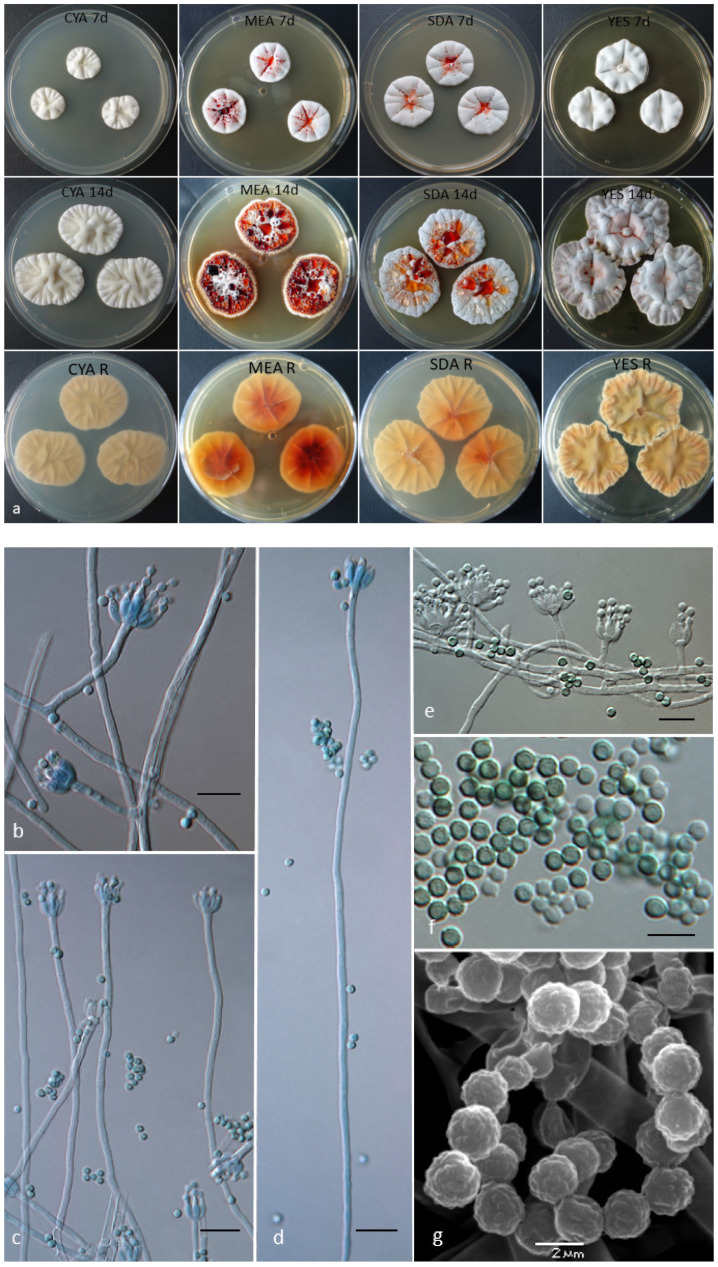
*Penicillium krskae* BiMM-F280. (**a**) Colonies on CYA, MEA, SDA, and YES; rows, from top to bottom: obverses after 7 and 14 days, and reverses after 14 days at 25 °C. (**b**–**e**) Conidiophores with conidia (on MEA, after seven days); (**f**) conidia (on MEA, after 14 days); (**g**) scanning electron microscopy (SEM) of conidia (on MEA, 20 days). Scale bars = 10 µm (**b**–**e**), 5 µm (**f**), 2 µm (**g**).

**Figure 2 jof-07-00557-f002:**
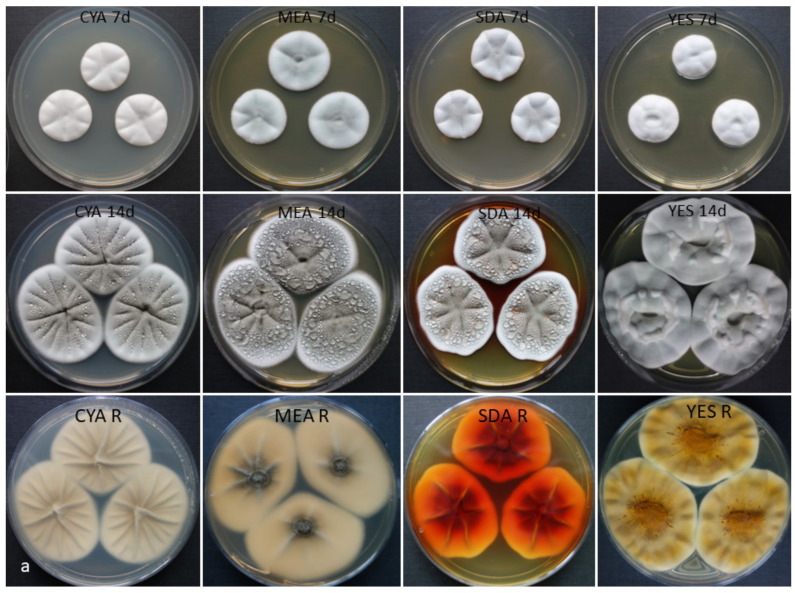
*Penicillium silybi* G-85. (**a**) Colonies on CYA, MEA, SDA, and YES; rows, from top to bottom: obverses after seven and 14 days, and reverses after 14 days at 25 °C. (**b**–**d**) Conidiophores with conidia (on MEA, after 25 days); (**e**) conidia (on MEA, after 14 days); (**f**) scanning electron microscopy (SEM) of conidia (on MEA, 20 days). Scale bars = 10 µm (**b**,**c**), 5 µm (**d**,**e**), 2 µm (**f**).

**Figure 3 jof-07-00557-f003:**
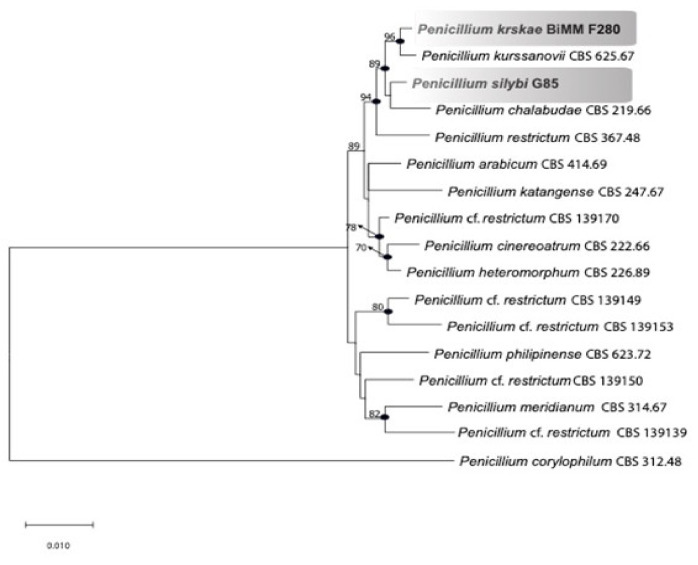
Maximum likelihood tree based on combined *BenA*, *CaM*, and *RPB2* of *P. restrictum*-clade within *Penicillium* section *Exilicaulis*. Bootstrap values higher than 70 are given above the branches. *Penicillium corylophilum* CBS 312.48^T^ was used as an outgroup. Names in bold are new species described in this study. Thickened nodes indicate Bayesian posterior probabilities ≥ 95%.

**Figure 4 jof-07-00557-f004:**
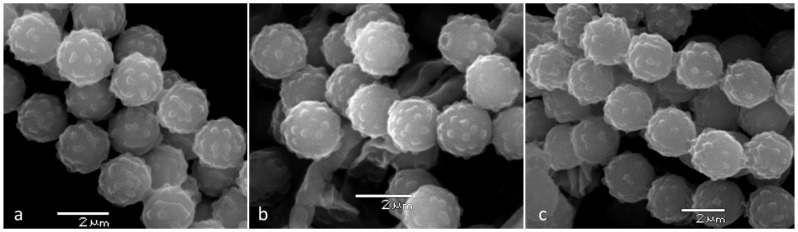
Scanning electron microscopy (SEM) of conidia (on MEA, 20 days) showing tuberculate ornamentation in (**a**) *Penicillium chalabudae* CBS 219.66 and (**b**) *P. kurssanovii* CBS 625.67, and aculeate-echinulate ornamentation in (**c**) *Penicillium restrictum* CBS 367.48.

**Figure 5 jof-07-00557-f005:**
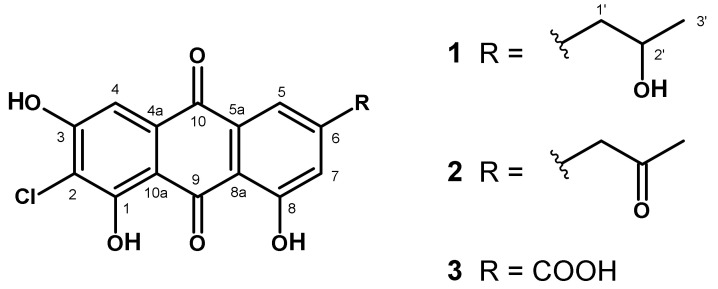
Structures of the new chloroemodines **1** and **2** and 2-chloroemodic acid (**3**).

**Table 1 jof-07-00557-t001:** List of fungal strains.

Species Name	Strain Numbers	Origin	GenBank Accession Numbers			
			ITS	BenA	CaM	RPB2
***P. krskae***	**BiMM-F280 ^a^**	Lab contaminant, Austria; ex-type	**MW794123**	**MW774594**	**MW774595**	**MW774593**
*P. silybi*	G85	Asymptomatic plant (*Silybum marianum*), USA; ex-type	KF367458	**MW774592**	**MW774591**	AB860248
*P. arabicum*	CBS 414.69	Soil, Syria; ex-type	KC411758	KP016750	KP016770	KP064574
*P. chalabudae*	CBS 219.66	Soil, Ukraine; ex-type	KP016811	KP016748	KP016767	KP064572
*P. cinereoatrum*	CBS 222.66	Soil, Ukraine; ex-type	KC411700	KJ834442	KP125335	JN406608
*P. corylophilum*	CBS 312.48	Unknown; ex-type	AF033450	JX141042	KP016780	KP064631
*P. heteromorphum*	CBS 226.89	Soil, China; ex-type	KC411702	KJ834455	KP016786	JN406605
*P. katangense*	CBS 247.67	Soil, DR Congo; ex-type	AF033458	KP016757	KP016788	KP064646
*P. kurssanovii*	CBS 625.67	Soil, Ukraine; ex-type	EF422849	KP016758	KP016789	KP064647
*P. meridianum*	CBS 314.67	Soil, South Africa; ex-type	AF033451	KJ834472	KP016794	JN406576
*P. philippinense*	CBS 623.72	Soil, Philippines; ex-type	KC411770	KJ834482	KP016799	JN406543
*P. restrictum*	CBS 367.48	Soil, Honduras; ex-type	AF033457	KJ834486	KP016803	JN121506
*P.* cf. *restrictum*	CBS 139139	Soil, South Africa; fynbos	JX140930	JX141055	JX157419	KP064607
*P.* cf. *restrictum*	CBS 139149	Soil, South Africa; fynbos	KP016819	JX141060	JX157467	KP064609
*P.* cf. *restrictum*	CBS 139150	Soil, South Africa; fynbos	KP016816	JX141061	JX157468	KP064610
*P.* cf. *restrictum*	CBS 139153	Soil, South Africa; fynbos	KP016818	JX141064	JX157474	KP064612
*P.* cf. *restrictum*	CBS 139170	Soil, South Africa; fynbos	-	JX141056	JX157420	KP064644

*P* = *Penicillium*, *^a^* BiMM, Bioactive Microbial Metabolites Unit, UFT-Tulln, Austria; CBS, Westerdijk Fungal Biodiversity Centre, Utrecht, the Netherlands; BiMM-F280 = CCF 6561, G-85 = CCF 6562; *P. cf restrictum*, fynbos strains that still require taxonomic treatment. Newly obtained data are in bold.

**Table 2 jof-07-00557-t002:** Comparison of the main distinctive phenotypic characteristics of phylogenetically related *Penicillium* spp. in series *Restricta*.

Fungus	Growth on CYA ^a^				Growth on MEA-CX250 ^b^	Growth and Acid Production on CREA ^c^	Red Exudate or Pigment Present	Conidial Morphology		
	15 °C	25 °C	30 °C	37 °C				Average Size in µm	Surface Ornamentation	
									LM	SEM
*P. krskae*	0	18–20	24–26	8–10	5 (15)	12–14 / +	Yes	2.1	sm-r-ro	verruc
*P. silybi*	5–7	26–28	25–27	3–5	10 (25)	20–22 /++	Yes **	2.0	ro-str	lob-retic
*P. chalabudae*	5–7	13–16	18–20	10–12	0 *	8–10 / ++	No	2.1	ro	tuber
*P. kurssanovii*	3–5	12–15	15–18	0	0 *	8–10 / - (±)	No	2.3	ro	tuber
*P. restrictum*	2–4	18–20	22–24	8–10	0 *	9–11 /- (±)	No	2.4	ro-ech	acul-ech

**^a^** All growth parameters in mm (diam.); **^b^** MEA with 250 ppm cycloheximide after 7 (and 14) d; * no spore germination even after 14 days; acid production: **^c^** ++ acid surrounding colony, + acid under the colony only, ± very weak acid production, - no acid production; ** red exudate and pigment produced on SGA, MEA with cycloheximide, or after prolonged cultivation on YES (14–16 d); LM: light microscope; SEM: scanning electron microscope; sm: smooth, r: finely roughened, ro: rough, str: striate, ech: echinulate (spinulose), verruc: verrucose, lob-retic: lobate-reticulate, tuber: tuberculate, acul-ech: aculeate-echinulate.

**Table 3 jof-07-00557-t003:** Growth response of related *Penicillium* species in series *Restricta* to cycloheximide in malt extract agar (MEA-CX) after 7 and 14 days incubation at 25 °C, relative to control with no cycloheximide (MEA).

Fungus	MEA Mm *	MEA-CX in mm (% Inhibition)		
		100	250	500
*P. krskae*	20–30	8 (60)–20 (33)	5 (75)–15 (50)	0 (100)–3 (90)
*P. silybi*	28–48	18 (36)–35 (27)	12 (57)–25 (48)	6 (78)–18 (62)
*P. chalabudae*	18–40	2 (89)–18 (55)	0 (100)–0 (100)	0 (100)–0 (100)
*P. kurssanovii*	22–42	0 (100)–0 (100)	0 (100)–0 (100)	0 (100)–0 (100)
*P. restricum*	18–30	0 (100)–5 (83)	0 (100)–0 (100)	0 (100)–0 (100)

MEA-CX100/250/500: MEA+ 100, 250, and 500 ppm cycloheximide; * colonies diameter in mm after 7 and 14 days at 25 °C, in the dark.

**Table 4 jof-07-00557-t004:** Metabolic composition of red exudate produced by two novel *Penicillium* species on MEA and PDA after 7 days of cultivation at 25 °C, as identified by LC–MS (El-Elimat et al., 2013, Paguigan et al., 2017).

Compound Name ^a^	Molecular Formula	*P. krskae*		*P. silybi*	
		PDA	MEA	PDA	MEA
2-hydroxyemodic acid	C_15_H_8_O_8_	−	−	+	−
(+)-2’S-Isorhodoptilometrin	C_17_H_14_O_6_	+	+	+	+
1’-Hydroxy-2’-ketoisorhodoptilometrin	C_17_H_12_O_7_	+	−	+	+
7-Chloro-1’-hydroxyisorhodoptilometrin	C_17_H_13_O_7_Cl	+ ^b^	+	−	+
1’-Hydroxyisorhodoptilometrin	C_17_H_14_O_7_	+ ^b^	+	+	+
2-Chloroemodic acid	C_15_H_7_ClO_7_	+ ^b^	+	+	+
2-Chloro-isorhodoptilometrin *	C_17_H_13_O_6_Cl	+ ^b^	+	−	+
2-Chloro-desmethyl dermoquinone *	C_17_H_11_O_6_Cl	+ ^b^	+	−	+
7-Chlorocitreosein	C_15_H_9_ClO_6_	+ ^b^	+	−	+
O-Demethyldermoquinone	C_17_H_12_O_6_	+	+	+	+
Emodic acid	C_15_H_8_O_7_	+	+	+	+
Emodin	C_15_H_10_O_5_	+	−	+	+
ω-Hydroxyemodin (citreorosein)	C_15_H_10_O_6_	+ ^b^	+	+	+

^a^ In alphabetical order; * newly discovered compound; ^b^ elucidated also by nuclear magnetic resonance (NMR).

**Table 5 jof-07-00557-t005:** Metabolite profile of five phylogenetically close *Penicillium* species (ex-type cultures) in series *Restricta* growing on yeast extract medium (YES) for 14 days at 25 °C and in the dark, after extraction with acetonitrile/water/acetic acid (79:20:1) and being measured by LC–MS/MS (concentration in µg.g^−1^).

Compound ^a^	*P. krsk*	*P. sil* ^b^	*P. chal*	*P. kurs*	*P. rest*
15-Hydroxyculmorin	-	-	-	-	1.10 *
7-Hydroxypestalotin	68.39	8.69	0.95	-	-
Chlorocitreorosein	0.54	355.20	0.12	0.19	0.08
Citreorosein	0.49	751.35	0.48	1.11	0.62
Emodin	-	36.02	-	0.01	0.03
Endocrocin	-	93.71 *	-	-	-
Hydroxysydonic acid	-	-	-	-	5.83 *
Iso-rhodoptilometrin	0.16	264.90	0.20	0.26	0.22
NP1243	1.67	5.66	0.28	0.62	0.18
Oxyskyrin	-	0.05*	-	-	-
Paxilline	4.31 *	-	-	-	-
Pestalotin	1.26	0.15	0.26	-	-
Skyrin	-	0.15 *	-	-	-
Tryptophol	-	-	1.52 *	-	-
Methylorsellinic acid	-	-	-	-	0.30 *

^a^ In alphabetical order; *P. krsk*: *P. krskae* BiMM-F280, *P. sil*: *P. silybi* G85; *P. chal*: *P. chalabudae* CBS 219.66; *P. kurs*: *P. kurssanovii* CBS 625.67; *P. rest*: *P. restrictum* CBS 367.48; ^b^ colonies with red pigment exuding into the agar; - under limit of detection (LOD); * compounds unique for the species.

## Data Availability

The datasets presented in this study can be found in online repositories. The names of the repositories and accession numbers can be found in the article and in the Appendix A. The data analyzed in this study are also available from the corresponding author on reasonable request.

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
