# Peer review of "Polyphasic Approach Utilized for the Identification of Two New Toxigenic Members of Penicillium Section Exilicaulis, P. krskae and P. silybi spp. nov."

_jof, 2021, doi:10.3390/jof7070557_

Round 1
Reviewer 1 Report
Title: Polyphasic approach utilized for the identification of two new toxigenic members of Penicillium Section Exilicaulis, P. krskae sp. nov. and P. silybi sp. nov.
Authors: R. Labuda, M. Bacher, T. Rosenau, E. Gasparotto, H. Gratzl, M. Doppler, M. Sulyok, A. Kubátová, H. Berger, K. Cank, H. A. Raja, N. H. Oberlies, C. Schüller, J. Strauss
Reference: jof-1294679
Article type: Research
Reviewer Comments:
The manuscript jof-1294679, entitled “Polyphasic Approach Utilized for the Identification of Two New Toxigenic Members of Penicillium Section Exilicaulis, P. krskae sp. nov. and P. silybi sp. nov.”, describe two new species, belonging to section Exilicaulis in series Restricta with phylogenetic affiliation to P. restrictum sensu stricto, using a polyphasic approach consisting of phenotypic, physiological, chemotaxonomic and multilocus phylogenetic (ITS, RPB2, benA, and CaM).
General comments:
- Text formatting must be revised, namely, section headings present dissimilar formatting throughout the text
- The English language can be improved in terms of writing style, spelling, and consistency, namely concerning:
- English language (please see the Specific comments section)
- Several italics are missing (please see the Specific comments section)
- Several typos have been detected
- Once an acronym has been defined, it should be used in the following text
- Use of non-scientific language
- Be consistent in the use of the Oxford comma
- Numbers lower than 10, must be written in full
- Avoid the use of very similar words in close proximity (e.g., line 183: “Bayesian Inference phylogenies were inferred”)
- Be consistent with the taxonomical concepts. Sometimes the authors refer to Section Aspergilloides, others to Subgenus Aspergilloides(line 65)
Specific comments:
Lines 28-30: please consider replacing “Abstract: Two new species, Penicillium krskae isolated from air as a lab contaminant in Tulln (Austria, EU) and Penicillium silybi isolated as an endophyte from asymptomatic milk thistle (Silybum marianum) stems in Josephine County (Oregon, USA) respectively, are described and illustrated.” with “Abstract: Two new species, Penicillium krskae [isolated from the air as a lab contaminant in Tulln (Austria, EU)] and Penicillium silybi [isolated as an endophyte from asymptomatic milk thistle (Silybum marianum) stems from Josephine County (Oregon, USA)] are described.”
Lines 38-39: please consider replacing “acids and other interesting bioactive extrolites, such as endocrocin, paxilline, pestalotin and 7-hydroxypestalotin.” with “acids and other interesting bioactive extrolites (i.e., endocrocin, paxilline, pestalotin, and 7-hydroxypestalotin).”
Lines 40-42: please consider replacing “Two new chloroemodic acid derivatives (2-chloro-isorhodoptilometrin, 2-chloro-desmethyldermoquinone) isolated from the exudate of P. krskae ex-type culture are elucidated by NMR and LC-MS and described herein.” with “Two new chloroemodic acid derivatives (2-chloro-isorhodoptilometrin and 2-chloro-desmethyldermoquinone) isolated from the exudate of P. krskae ex-type culture are analyzed by Nuclear magnetic resonance (NMR) and Liquid chromatography–mass spectrometry (LC–MS).”
Line 48: please consider replacing “>483 species.” with “more than 483 species”
Line 51: please consider replacing “species are found from myriad of ecological habitats” with “species have been isolated from a myriad of ecological habitats”
Line 54: are these stains used in the pharmaceutic industry?
Line 77: please consider replacing “extrolites examination for identifying” with “extrolites examination to identify”
Line 108: please consider replacing “were noted after seven” with “were observed after seven”
Lines 100-110: please consider replacing “on all media in order to observe and record changes in pigmentation of the colonies” with “on all media to observe and record changes in colonies pigmentation”
Line 116: please consider replacing “and MEA-CX500, resp.)” with “and MEA-CX500, respectively)”
Lines 117-118: please consider removing the sentence “Due to its thermal instability at 119-120°C, cycloheximide was filter sterilized and added to the medium after autoclaving (at approx. 70 °C).”
Line 116: please consider replacing “MEA was used” with “MEA were used”
Lines 147 - 151: please consider replacing “DNA was extracted from the strains grown on MEA for 7 days using the DNeasy Plant Minikit (Qiagen, Germany). Amplification of the internal transcribed spacer (ITS), RNA-polymerase II second largest subunit (RPB2), partial tubulin (BenA), and calmodulin (CaM) were performed with primers (Sigma-Aldrich, USA) using slightly adjusted amplification procedures as described by Visagie et al., (2014) [19].” with “DNA was extracted from the strains grown on MEA for seven days using the DNeasy Plant Minikit (Qiagen, Germany). Amplification of the ITS, RPB2, BenA, and CaM were performed as previously described with modifications [19].”
Lines 152-158: consider replacing this text into a table resuming all PCR conditions and indicating primer sequence and bibliographic reference.
Lines 163-164: please consider removing the sentence “This table also provides GenBank accession numbers to ITS, BenA, CaM and RPB2 sequences for all accepted species in Penicillium restrictum-clade within the sect. Exilicaulis.”
Lines 166-167: please consider replacing “Nucleotide sequences from 17 Penicillium species (Table 1) for the genes BenA, calmodulin and RNA polymerase II second largest subunit were aligned using ClustalW” with “Nucleotide sequences from 17 Penicillium species (Table 1) for the genes BenA, CaM and RPB2 were aligned using ClustalW”
Lines 205-206: please consider replacing “the plates were checked for their purity” with “the plates were verified for the presence of contamination”
Line 212: please consider removing the text “, i.e. colony and agar under the colony,”
Line 303: please consider removing the text “, a world authority on the analytics of mycotoxins.”
Line 304: please consider replacing “Colonies on CYA slowly to moderately growing, 18-20 mm diameter after 7 d at 25°C,” with “Colonies on CYA presented slow to moderate growth with 18-20 mm diameter, after seven days, at 25°C,”. Please consider this comment and apply it throughout the remaining text.
Lines 304 - 343: consider replacing this text into two tables resuming all colony, hyphae, conidiophores, phialides, and conidia characteristic
Lines 345 - 352: convert into a text (not merely an enumeration of compounds)
Line 309: the italics formatting is missing in Penicillium
Lines 408 - 416: convert into a text (not merely an enumeration of compounds)
Lines 440 - 441: please consider replacing “., respectively. The tree with the highest log likelihood (-4458.29) is shown.” with “., respectively (Figure 3).”
Line 442: please consider replacing “Penicillium krskae is phylogenetically closest to P. kurssanovii Chalab., while P. silybi is sister to P. chalabudae Visagie” with “Penicillium krskae is phylogenetically close to P. kurssanovii Chalab., and P. silybi to P. chalabudae Visagie”
Line 445: please consider replacing “series Restricta (formerly the P. restrictum-clade)” with “series Restricta (formerly P. restrictum-clade)”
Line 449: please consider replacing “Maximum Likelihood tree” with “Maximum likelihood tree”
Lines 459 - 461: please consider replacing “section), and (5) conidial morphology. Compilation of the main distinguishing phenotypic characteristics of the two new species compared with the other phylogenetically closest relatives within the P. restrictum-clade is listed in Table 2.” with “section), and (5) conidial morphology (Table 2).”
Lines 462 - 463: please consider replacing “(1) Growth at 37°C. Out of five species investigated in this study, only P. kurssanovii did not grow at 37°C (CYA, MEA). The slowest growth at this temperature was noted for” with “(1) Growth conditions: At 37ºC only P. kurssanovii did not present any growth (CYA, MEA). The slowest growth at this temperature was observed for”
Lines 465 - 470: please consider replacing “Growth at 25°C. P. silybi showed the fastest colony growth on CYA (26-28 mm in diam) and on MEA (28- 30 mm in diam) after 7 days when compared to those of other four species in the investigated group, ranging from 12-20 mm in diam (CYA) and 17-22 mm in diam (MEA). No growth with no spore germination at 15 °C on CYA was detected for P. krskae. The other four species showed at least minimal growth ranging from 2 to 7 mm in diam. at this temperature.” with “At 25°C, P. silybi presented the fastest growth on CYA (26-28 mm) and MEA (28-30 mm), after seven days when compared the other four species in this group, ranging from 12-20 mm (CYA) and 17-22 mm (MEA). At 15ºC, P. krskae presented no growth nor spore germination on CYA. The remaining four species presented minimal growth, ranging from 2 to 7 mm. ”
Lines 472 - 473: please consider replacing “(2) Response to cycloheximide. Growth response of the new species along with the phylogenetically closest P. chalabudae” with “(2) Response to cycloheximide: Growth response of the new species and the phylogenetically close P. chalabudae”
Line 475: please consider replacing “P. silybi possessed a moderate growth” with “P. silybi presented moderate growth”
Line 477: please consider replacing “(3) Growth and acid production on CREA. In general, all species showed” with “(3) Growth and acid production on CREA: In general, all species showed”
Lines 481 - 485: confusing sentence, please consider rephrasing
Line 477: please consider replacing “(5) Conidia morphology. Detailed study of the morphology of conidia in terms of size” with “(5) Conidia morphology: Detailed study of conidia morphology in terms of size”
Lines 488 - 492: please consider replacing “The spore surface ornamentation varied among the investigated species, as well. Scanning electron microscopy (SEM) revealed four morphs, namely verrucose in P. krskae (Figure 1g), lobate-reticulate in P. silybi (Figure 2f), tuberculate in P. chalabudae (Figure 4a) and P. kurssanovii (Figure 4b), and aculeate-echinulate in P. restrictum (Figure 5).” with “Spore surface ornamentation varied from verrucose (P. krskae, Figure 1g), lobate-reticulate (P. silybi, Figure 2f), tuberculate (P. chalabudae, Figure 4a and P. kurssanovii, Figure 4b), and aculeate-echinulate (P. restrictum, Figure 5).”
Line 506: please consider replacing “Penicillium chalabudae CBS 219.66 and (b) P. kurssanovii CBS 625.67.” with “P. chalabudae CBS 219.66 and (b) P. kurssanovii CBS 625.67.” Also, please verify the use of italics
Lines 507 - 508: please consider moving the text to below the table
Lines 520 - 521: not a result, consider moving the text to a more adequate section
Line 529: please consider replacing “Metabolic Profile of the Related Penicillium Species in Series Restricta” with “Metabolic Profile of the Related Penicillium Species in Series Restricta”
Lines 530 - 532: please consider removing the sentence “The metabolic profile of the five closely related species (growing on YES medium for 14 days, at 25°C, in dark) without any conspicuous exudate production is highlighted in Table 5.”
Lines 530 - 533: not a result, consider moving the text to a more adequate section
Line 535: please consider replacing “were found in common for all five ex-type cultures” with “were commonly found in the five ex-type cultures”
Lines 538 - 533: please consider replacing “were found only in the extract of P. restrictum. Endocrocin and two bianthraquinones (oxyskyrin and skyrin) were detected only in P. silybi, and production of tremorgenic paxilline was observed exclusively in P. krskae. Similarly, tryptophol was found only in the extract of P. chalabudae.” with “were only found in P. restrictum, endocrocin and two bianthraquinones (oxyskyrin and skyrin), paxilline in P. silybi, and tryptophol in P. chalabudae.”
Lines 549 - 550: please consider replacing “Compounds 1 and 2 could only be isolated as a mixture with 2-chloroemodic acid (3) as its main component (1:2:3 = 20:5:75). ” with “Compounds 1 and 2 were only extracted using a 2-chloroemodic acid based mixture (3). ” Is (3) a bibliograpphic reference? If so, it is not according to the journal guidelines.
Lines 555 - 557: not a result, consider moving the text to a more adequate section
Lines 572 - 573: please consider replacing “confirm structure 2 as drawn in Figure 6. Compounds 1 and 2 represent hitherto undescribed” with “confirm structure 2 (Figure 6). Compounds 1 and 2 represent undescribed”
Lines 592-593: please consider replacing “P. parvum-and P. restrictum- , clades, ” with “P. parvum- and P. restrictum-clades, ”
Line 535: please consider replacing “closest species P. kurssanovi,” with “close species P. kurssanovi,”
Line 536: please consider replacing “(i.e. P. arabicum, P. katangense)” with “(i.e., P. arabicum and P. katangense)”
Lines 607-608: please consider rephrasing the text “that they are good and distinct species”
Line 609: please consider replacing “adopted in the P. restrictum-clade” with “adopted for the P. restrictum-clade”
Line 611: please consider removing the text “(in alphabetical order),”
Lines 614-615: please consider rephrasing the text “The authors noted that there is still need for a comprehensive revision ”
Line 623: please consider replacing “Additionally, he” with “Additionally, this author”
Lines 624-625: please consider replacing “two series (ser. Restricta and Citreonigra) on the basis of growth rates at 25°C and stipe length, and accepted 22 species” with “two series (Restricta and Citreonigra) based on growth rates at 25°C and stipe length, accepting 22 species”
Line 626: please consider removing the text “with the application of molecular phylogeny”
Line 628: please consider removing the text “ that there are as many as”
Line 631: please consider replacing “traits and profiles of secondary metabolites,” with “traits and secondary metabolites profiles,”
Line 636: please consider removing the text “under SEM”
Line 642: please consider replacing “tool for distinguishing phylogenetically closest taxa” with “tool to distinguish phylogenetically close taxa”
Line 648: make a paragraph before “The new species”
Line 648: please consider replacing “the phylogenetically closest P.” with “the phylogenetically close P.”
Line 651: please consider replacing “cose vs. tuberculate under SEM as well as an average size (2.1 vs. 2.3 µm),” with “cose vs. tuberculate and an average size (2.1 vs. 2.3 µm),”
Line 655: please consider replacing “two species aside, as P. krskae produces” with “two species, as P. krskae produces”
Line 667: please consider removing the text “in laboratory experiments”
Line 672: please consider removing the text “Over the years”
Lines 677-678: please consider replacing “In this paper” with “In the present study”
Lines 679-680: please consider removing the text “in laboratory experiments” using positive and negative modes and comparison to in house standards”
Line 695: italics missing in Staphylococcus aureus and in vitro
Lines 697-698: please consider removing the text “was undertaken, which ”
Line 705: please consider replacing “Overall, our study demonstrates” with “Overall, the present study demonstrates”
Figures:
Consider joining figures 4 and 5 in a single figure.
Scientific comments:
Material and Methods:
- Some subsections within the Material and Methods are unnecessarily too descriptive. Please consider simplifying the text.
- If the amplification protocol different from the protocol described in Visagie et al., (2014), the adjustments must be described.
Author Response
Comments to reviewers for manuscript JoF_1294679
entitled “Polyphasic approach utilized for the identification of two new toxigenic members of Penicillium Section Exilicaulis, P. krskae sp. nov. and P. silybi sp. nov. “
by R. Labuda, M. Bacher, T. Rosenau, E. Gasparotto, H. Gratzl, M. Doppler, M. Sulyok, A. Kubátová, H. Berger, K. Cank, H. A. Raja, N. H. Oberlies, C. Schüller, J. Strauss
In general:
We adopted nearly all changes as suggested by both reviewers:
If text is added, the it is indicated ….. text text text
If text is removed/skipped, then it is indicated as ….. text text text
R: Lines 152-158: consider replacing this text into a table resuming all PCR conditions and indicating primer sequence and bibliographic reference.
Complete set of primers used as well as the thermal cycles are well described (in form of tables) available in reference nr 19. (Visagie et al., 2014, Study in Mycology 78). The following has been inserted into text.
Regions of internal transcribed spacer (ITS), β-tubuline (BenA), calmodulin (CaM) and RNA-polymerase II second largest subunit (RPB2) were amplified using primer pair ITS1 and ITS4, Bt2a and Bt2b, CMD5 and CMD6, and 5Feur and 7CReur, respectively (19). The PCR thermal cycle programs used for amplification were those as described in (19).
R: Lines 304 - 343: consider replacing this text into two tables resuming all colony, hyphae, conidiophores, phialides, and conidia characteristic
A formal description of fungi (including new species) is always in the form of text.
R: Lines 530 - 532: please consider removing the sentence “The metabolic profile of the five closely related species (growing on YES medium for 14 days, at 25°C, in dark) without any conspicuous exudate production is highlighted in Table 5.”
This sentence emphasis that the metabolites were not extracted from an exudate (droplets).
R: Lines 549 - 550: “Compounds 1 and 2 were only extracted using a 2-chloroemodic acid based mixture (3). ” Is (3) a bibliograpphic reference? If so, it is not according to the journal guidelines.
(3) in bold indicates the number of substance, not a reference
R: Lines 555 - 557: not a result, consider moving the text to a more adequate section
This sentence is a consequence (result) of the unchanged H-4 shift leading over to the description of the elucidation of the side chain, where the differences of the derivatives are located. Well, it is a result.
R: Figures: Consider joining figures 4 and 5 in a single figure.
It has been done!
Reviewer 2 Report
Here is the review of the paper entitled "Polyphasic Approach Utilized for the Identification of Two New Toxigenic Members of Penicillium Section Exilicaulis, P. krskae sp. nov. and P. silybi sp. nov." written by Roman Labuda & co-authors.
The paper presents scientific descriptions of two fungal species new to science: Penicillium krskae and P. silybi. Both species are well separated from all described Penicillium species by phenotypic (conidial ornamentation, production of red exudate and red pigments), physiological (growth at 37°C, response to cycloheximide and CREA), and chemotaxonomic traits (production of specific extrolites). Multilocus phylogenetic analysis based on combination of RPB2, benA, and CaM marker genes placed both taxa in series Restricta of the section Exilicaulis, close P. restrictum sensu stricto. New species are known only from one isolate (asexual state). Both are thoroughly described in the mansuscript following to the newest version of International code of nomenclature for algae, fungi and plants. Both species produce a large number of toxic anthraquinoid pigments and interesting bioactive extrolites (e.g. endocrocin, paxilline, pestalotin and 7-hydroxypestalotin). Culture extract of P. silybi is caharacterized by the presence of two bianthraquinones, (i.e. skyrin and oxyskyrin), while two new chloroemodic acid derivatives (2-chloro-isorhodoptilometrin, 2-chloro-desmethyldermoquinone) were isolated from the exudate of P. krskae.
The research methods used in the paper are thorough, suitable, and well conducted. The topic is very interesting for JoF audience. There ar only a few suggested corrections/additions included in the attached review of the manuscript file.
After minor revision the paper deserves to be published in JoF.
Best,
Reviewer

Author Response

(The authors gave the same response as above.)
